# Comparative Analysis of mRNA and miRNA Expression between Dermal Papilla Cells and Hair Matrix Cells of Hair Follicles in Yak

**DOI:** 10.3390/cells11243985

**Published:** 2022-12-09

**Authors:** Xiaolan Zhang, Pengjia Bao, Qingbo Zheng, Min Chu, Chunnian Liang, Xian Guo, Xiaoyun Wu, Meilan He, Chengfang Pei, Ping Yan

**Affiliations:** 1Key Laboratory of Animal Genetics and Breeding on Tibetan Plateau, Ministry of Agriculture and Rural Affairs, Lanzhou Institute of Husbandry and Pharmaceutical Sciences, Chinese Academy of Agricultural Sciences, Lanzhou 730050, China; 2Animal Husbandry Technology Extending Station of Tianzhu Tibetan Autonomous County, Tianzhu 733200, China

**Keywords:** hair follicles, yak, dermal papilla cells, hair matrix cells, miRNAs

## Abstract

The interaction between the dermal papilla cells (DPCs) and epidermal hair matrix cells (HMCs) of hair follicles (HFs) is crucial for the growth and development of HFs, but the molecular mechanism is complex and remains unclear. MicroRNAs (miRNAs) are the key signaling molecules for cellular communication. In this study, the DPCs and HMCs of yak were isolated and cultured, and the differentially expressed mRNA and miRNA were characterized to analyze the molecular basis of the interaction between DPCs and HMCs during hair follicle (HF) development in yak. The mRNA differential expression and functional enrichment analysis revealed that there were significant differences between DPCs and HMCs, and they showed the molecular functional characteristics of dermal cells and epidermal cells, respectively. Multiple KEGG pathways related to HF development were enriched in the highly expressed genes in DPCs, while the pathways associated with microbiota and immunity were significantly enriched in the highly expressed genes in HMCs. By combining analysis with our previous 10× genomics single-cell transcriptome data, 39 marker genes of DPCs of yak were identified. A total of 123 relatively specifically expressed miRNAs were screened; among these, the miRNAs associated with HF development such as miR-143, miR-214, miR-125b, miR-31, and miR-200 were presented. In conclusion, the large changes in yak DPCs and HMCs for both mRNA and miRNA expression were revealed, and numerous specifically expressed mRNAs and miRNAs in DPCs or HMCs were identified, which may contribute to the interaction and cellular communication between DPCs and HMCs during HF development in yak.

## 1. Introduction

A hair follicle is a complex mini-organ that is characterized with continual self-renewal in mammals after birth. Dermal and epidermal cell lineages, which are composed of various cell types, are the main components of hair follicles and orchestrate the dynamic changes of hair growth [1,2]. The intercellular communications between dermal papilla cells (DPCs) and surrounding epithelial hair matrix cells (HMCs) are important for hair follicle development, which promotes the regeneration and shedding of hair shafts. At anagen onset, HMCs, also called transit-amplifying cells that are derived from hair follicle stem cells (HFSCs) located in the bulge region of hair follicles, are activated and begin to proliferate rapidly in response to the signals from DPCs. In the following growth, DPCs are gradually enveloped by HMCs. At the same time, the signals from DPCs guide the differentiation of HMCs into different hair-follicle-cell lineages, including hair shaft (HS) and inner root sheath (IRS) [3,4,5]. Similarly, at catagen, the epithelial HMCs and outer root sheath cells in the hair bulb begin apoptosis by receiving the signals from DPCs and then the development of the hair follicle gradually enters telogen [6]. It is generally recognized that the dermis cells are the induction cells and the epidermis cells are the response cells during HF morphogenesis and cycling, but their signaling transduction and molecular interactions are intricate and still not well understood [7,8].

Numerous studies and methods have been used to investigate the key components of DPCs and HMCs and their interaction mechanism during HF development [9,10]. The Sonic hedgehog (SHH) signaling pathway was reported to be involved in the coordination of the development of epidermis and dermis in hair follicle morphogenesis [11]. The classical Wnt signaling pathway plays an important role during hair follicle cycle and hair follicle morphogenesis [12]. In addition, Notch signals were found to determine the location of epidermal cells and then regulate the formation and differentiation of epidermal cell lineage in hair follicle development [13]. In recent years, single-cell sequencing technology has provided unprecedented convenience for the determination of the heterogeneity of hair follicle cells and revelation of the epithelial–mesenchymal interactions [14,15]. However, whether the single-cell sequencing is based upon fluorescence-activated cell sorting (FACS) or 10× genomics single-cell sequencing, the reliability of cell clustering depends on the accuracy of known markers [5,15,16]; thus, the cell clustering of the single-cell sequencing data may be difficult in non-model animals. The combined analysis of sequencing data from cultured cells and 10× genomics single-cell data of hair follicles will effectively facilitate the accurate identification of key cell types and their marker genes.

MicroRNAs (miRNAs) are ~22 nt non-coding small RNA and with a high conservation across evolution in diverse eukaryotic lineages [17]. Accumulating evidence suggests that miRNAs can function in intercellular communication through the paracrine mechanism [18,19], for example, exosomes, which are important carriers involved in intercellular communication and released by different cells. It was reported that the mature miRNA accounts for 41.7% of the total exosomal RNA, and most of these miRNAs are considered key bioactive components of exosomes [20,21]. Recently, the miRNAs within DPCs were reported to regulate the proliferation and differentiation of HFSCs and are associated with the regulation of the HF cycle [22,23,24]. In fact, numerous miRNAs have been reported to play roles in regulating hair follicle development and hair loss disorders [25,26], but the regulatory mechanisms and which miRNAs are involved in intercellular communication between different cells remain unclear. The premise of intercellular communication is that molecules may be significantly differentially expressed between neighboring cells. Detection of differentially expressed miRNAs between DPCs and HMCs undoubtedly provides a simple and effective method for exploring the miRNAs involved in intercellular signal transmission.

Yak HFs may develop some unique molecular properties during long-term adaptation to the harsh alpine environment, and the seasonal development of yak HFs is of great significance in this regard [27,28]. The interaction between DPCs and HMCs regulated the regeneration and shedding of hair shafts. Although much research has been conducted on DPCs and HMCs, few studies have systematically compared the molecular differences between these two types of cells, and so far, no studies have reported hair follicle cells in yak. Thus, in the present study, the DPCs and HMCs of yak hair follicles were isolated and cultured, and RNA-seq was performed to detect their differentially expressed mRNAs and miRNAs. As a result, the functional properties and important signaling molecules of DPCs and HMCs were identified and analyzed. The findings of this study will provide effective data resources for revealing the molecular mechanism of the interaction between DPCs and HMCs and may provide new insight into the function of HFs in yak adaptation to the alpine environment.

## 2. Materials and Methods

### 2.1. Animal and Sample Collection

Tianzhu white yaks, roughly one week old, were selected for skin sample collection and hair follicle cell isolation. Briefly, after the hair in the scapulae region was cleaned using scissors, 2% lidocaine was injected, and the skin samples of yak were collected using a medical skin sampler. Veterinary penicillin and streptomycin powder were sprinkled on the wound, and a sterile compress was applied to prevent infection. The collected skin samples were sterilized with 75% alcohol and washed with phosphate-buffered saline containing 1% penicillin–streptomycin (D-PBS) then placed in D-PBS and immediately brought back to the laboratory for subsequent experiments. The skin samples used in histoimmunofluoresence staining were preserved in our laboratory during the previous study [28]. All the yaks were from Tianzhu country, Wuwei City, Gansu Province, China.

All the experimental procedures involved in this study were approved by the Animal Management and Ethics Committee of the Lanzhou Institute of Animal Science and Veterinary Medicine, Chinese Academy of Agricultural Sciences (Permit No. SYXK-2014-0002).

### 2.2. Isolation and Purification of DPCs and HMCs

The skin sample was sterilized with 75% alcohol, washed at least three times with D-PBS and cut into 0.5 × 0.3 cm strips on a clean bench. The strips of skin tissue were treated with 0.25% dispase II for 2 h at 37 °C, and the single HFs could be separated from the strips using microscopic tweezers under a stereomicroscope. Hair bulbs of the single hair follicles were cut off using a microscopic shear and placed in 6-well dishes. Then, 0.25% trypsin was added for further digestion for 30 min, followed by the slow addition of DMEM/F12 medium supplemented with 1% penicillin–streptomycin, 10 ng/mL IGF1 (Sigma, Burlington, MA, USA), 20 ng/mL EGF (Sigma, Burlington, MA, USA), and 10% FBS. After being cultured in a 37 °C and 5% CO_2_ atmosphere for about a week, it could be observed that the adherent cells migrated from the hair bulbs under the microscope. The medium was replaced every two days. The initially migrated cells included DPCs and HMCs, and a clear boundary existed between the two types of cells. The method of differential trypsin digestion was used to purify the cells. When the cells reached 70~80% confluence, after trypsinization for 2 min, the DPCs separated from the culture plate and were transferred to a culture flask for further culture. The remaining adherent cells were washed with PBS two times and subjected to further trypsinization for 5 min; then, the HMCs could be separated from the culture plate, and these purified HMCs were transferred to a new culture flask for further culture.

### 2.3. RNA Isolation, Library Construction, and Sequencing

The purified passage2 DPCs and HMCs were seeded in a 6 cm Petri dish with three repetitions of each cell type. Total RNA of DPCs and HMCs were extracted using Trizol reagent (Invitrogen, Carlsbad, CA, USA). RNA purity and quantification were evaluated using a NanoDrop 2000 spectrophotometer (Thermo Scientific, Waltham, MA, USA). RNA integrity was assessed using an Agilent 2100 Bioanalyzer (Agilent Technologies, Santa Clara, CA, USA). Transcriptome and small RNA sequencing were conducted by OE Biotech Co., Ltd. (Shanghai, China).

For mRNA sequencing, in total, six libraries from DPCs (*n* = 3) and HMCs (*n* = 3) were constructed using a TruSeq Stranded mRNA LT Sample Prep Kit (Illumina, San Diego, CA, USA) according to the manufacturer’s instructions. The libraries were sequenced on an Illumina HiSeq 4000 platform, and 150 bp paired-end reads were generated. Raw data (raw reads) of fastq format were firstly processed using Trimmomatic [29], and low-quality reads were removed to obtain the clean reads.

For miRNA sequencing, a total amount of 1 µg RNA per sample was used for the small RNA library construction and six libraries from DPCs (*n* = 3) and HMCs (*n* = 3) were constructed using TruSeq Small RNA Sample Prep Kits (Cat. No. RS-200-0012, Illumina, San Diego, CA, USA.) following the manufacturer’s recommendations. Briefly, total RNA of each sample was ligated to adapters at each end. Then, the adapter-ligated RNAs were reverse transcribed to cDNA and PCR amplification was performed. The PCR products ranging from 140 to 160 bp (the length of small noncoding RNA plus the 3′ and 5′ adaptors) were isolated and purified as small RNA libraries. Library quality was assessed on an Agilent Bioanalyzer 2100 system using high-sensitivity DNA chips. The libraries were finally sequenced using the Illumina HiSeq 2500 platform, and 50 bp single-end reads were generated. The basic reads were converted into raw reads by base calling. Low-quality reads were filtered, and the reads with 5′ primer contaminants and poly (A) were removed. The reads without the 3′ adapter and insert tag and the reads shorter than 15 nt or longer than 41 nt from the raw data were filtered to obtain the clean reads.

### 2.4. Transcriptome Assembly and mRNA Expression Analysis

The clean reads were mapped to the *Bos*_*grunniens* genome (Bosgru_v3.0, http://asia.ensembl.org/Bos_grunniens/Info/Index8, accessed on 8 June 2022) using HISAT2 [30]. The FPKM of each gene was calculated using Cufflinks [31] and the read counts of each gene were obtained by HTSeq-count [32]. Differential expression analysis was performed using the DESeq (2012) R package [33] and the thresholds of *q*-value < 0.05 and |log2FoldChange| > 1 were used for identifying the differentially expressed genes (DEGs). A volcano plot of DEGs was analyzed to demonstrate the expression distribution of genes in different groups. GO and KEGG [34] pathway enrichment analysis of DEGs was performed using the ClusterProfiler R package based on the hypergeometric distribution.

### 2.5. Quantification of miRNA Expression Level and Target Prediction

The length distribution statistics of clean reads were analyzed using the program written by OE Biotech Co., Ltd. (Shanghai, China). All the clean reads were mapped to the yak genome, and then subjected to the Bowtie_v1.3.1 [35] search against Rfam v.10.1 [36]. Non-coding RNAs were annotated as rRNAs, tRNAs, small nuclear RNAs (snRNAs), and so on. The known miRNAs were identified by aligning against the miRBase v22 database (http://www.mirbase.org/, accessed on 29 August 2022) [37] and then the unannotated reads were analyzed by mirdeep2 [38] to predict novel miRNAs. TPM was used for estimating miRNA expression levels [39]. Differentially expressed miRNAs between DPCs and HMCs were calculated using the DESeq R package [33] and filtered with the threshold of *q*-value < 0.05 and |log2FoldChange| > 1. The potential targets of miRNAs were predicted by MiRanda (http://www.microrna.org/microrna/home.do, accessed on 29 August 2022) and TargetScan (http://www.targetscan.org/vert_71/, accessed on 29 August 2022). The miRNA–mRNA interaction network was established and visualized using Cytoscape (v3.6.1).

### 2.6. Immunofluorescent Staining

For cell immunofluorescent staining, the cells were seeded in a 12-well plate. After 24 h, the cells were fixed with 4% paraformaldehyde followed by a permeabilization procedure in 0.5% TritonX-100 (Sorlabio, Beijing, China) for 10 min. The cells were blocked with 10% horse serum (Solarbio, Beijing, Chain) for 1 h at room temperature. Primary antibodies against goal proteins were then incubated at 4 °C overnight. Subsequently, Alexa Fluor^®^ 488 labeled Goat Anti-Rabbit IgG (Abcam, Cambridge, UK) was used to specifically bind to the primary antibody, DAPI (Sorlabio, Beijing, China) was used for nuclei staining and the pictures were taken using a confocal microscope (ZEISS, Jena, German). For histoimmunofluoresence, the skin samples in anagen were cut into 7 µm sections with a freezing microtome (Leica, Nussloch, Germany). The slides were fixed with 4% paraformaldehyde; the subsequent process was as described in the cell immunofluorescence section. The following information of primary antibodies was used: α-SMA (Abcam, Cambridge, UK, VIM (Bioss, Beijing, China), and ITGA6 (Proteintech, Wuhan, China).

### 2.7. qPCR

Cell RNAs were converted to cDNA using a PrimeScriptTM RT Kit with gDNA Eraser (Takara, Dalian, China) for the mRNA and pri-miRNA qPCR. The cDNA used for qPCR of miRNAs was synthesized using a 1st Strand cDNA Synthesis Kit (Takara, Dalian, China). The qPCR was performed using TB Green™ Premix Ex TaqTM II (Takara, Dalian, China) on a Bio-Rad CFX96 Touch™ Real Time PCR Detection System (Bio-Rad, Hercules, CA, USA). The procedure of qPCR used in quantifying mRNA and pri-miRNA was as follows: 95 °C for 3 min, followed by 40 cycles of 95 °C for 10 s and 60 °C for 30 s, and the expression abundance was normalized relative to that of *GAPDH*. The reactions of miRNA qPCR were incubated at 95 °C for 10 s, followed by 40 cycles of 95 °C for 5 s and 60 °C for 20 s. Primers used in qPCR were designed by oligo 6. The sequences of pri-miRNAs were obtained from the Ensembl database by extending the 5′ flanking and 3′ flanking sequences to 150 bp of the selected miRNAs. For miRNA qPCR detection, the mature sequence of miRNAs was used as the forward primer, and the reverse universal primer was provided by the miRNA cDNA Synthesis kit. U6 was used as the internal control in quantifying miRNAs. All the primers are listed in the Appendix A.

### 2.8. Statistical Analysis

The data were analyzed using the 2^−∆∆CT^ method. The qPCR results are presented as mean ± standard error (SEM). Histograms in the present study were analyzed with GraphPad Prism 8.0 software (GraphPad Software, San Diego, CA, USA) with the *t*-test. Statistical significance is presented as * *p* < 0.05, ** *p* < 0.01, and *** *p* < 0.001.

## 3. Results

### 3.1. Isolation of DPCs and HMCs from Hair Follicles of Yak

The hair bulb is mainly composed of DPCs and HMCs closely surrounding DPCs. To acquire the cultured DPCs and HMCs of yak, the intact hair bulb regions from HFs were isolated and transferred into six-well culture dishes. After 5~7 days, the two types of cells could be observed to migrate out and grow adherently, and they differ markedly in morphology (overall schema: Figure 1a). The DPCs were mainly fusiform or triangular, while the HMCs were characterized by cobblestone-like cells and grew in the shape of paving stones. Moreover, the size of the HMC is smaller than that of the DPC, and there is a clear boundary between the two types of cells. Morphological characteristics of DPCs and HMCs isolated in this study are consistent with the previous studies [40,41,42]. Then, the method of differential trypsin digestion was used to obtain the purified DPCs and HMCs (Figure 1b). α-SMA is a well-known marker gene of DPCs [43,44] which was used for the identification of the DPCs. The immunofluorescence result showed that the α-SMA was highly expressed in the DPCs while not expressed in the cytoplasm of HMCs (Figure 1c), suggesting the DPCs were successfully identified. These results for isolated and purified DPCs and HMCs of yak hair follicles show that there were significant differences between DPCs and HMCs both in morphological characteristics and molecular expression, which may be the basis of the mesenchymal–epithelial interactions and signal exchange during hair follicle development.

### 3.2. Analysis of the Differentially Expressed Genes between DPCs and HMCs

To elucidate the molecular biological basis of DPCs and HMCs exerting their functions and interaction in hair follicle development, the mRNA and miRNA expression profiles of DPCs and HMCs were detected by RNA-seq. For mRNA sequencing, six cDNA libraries were constructed from DPC and HMC samples (*n* = 3 for each). The average of raw reads for these 6 libraries was 49.69 million reads. After discarding low-quality reads, an average of 48.74 million clean reads were retained with 7.0~7.37Gb of the clean base. The clean reads were mapped to the yak reference genome (Bosgru_v3.0, http://asia.ensembl.org/Bos_grunniens/Info/Index8, accessed on 8 June 2022) using HISAT2 [30]. As a result, 92.04~93.0% of the clean reads from all the samples were mapped to the reference genome (Appendix A).

A total of 15,646 genes were identified as being expressed in at least one of the six samples based on the known reference gene database and annotation file, and the distribution of FPKM values for these genes is shown in Figure 2a. The result showed that the expression patterns of the genes were similar among all samples, the highest number of genes were expressed at the level of FPKM ≥ 10, and more than 70% of the genes were expressed at the level of FPKM > 1. The analysis result of the correlation coefficient between samples showed that the samples in the same group with a high correlation (>0.99) and the two types of cells were clearly separated by the correlation coefficient (Figure 2b).

Then, the differentially expressed genes (DEGs) between DPCs and HMCs were analyzed with a threshold of adjusted *p*-value < 0.05 and |log2FC| > 1, and a total of 4600 DEGs were identified. Among these DEGs, 2265 mRNAs were upregulated, and 2335 mRNAs were downregulated in DPCs compared with HMCs (Figure 2c). It could be found that the number of DEGs between DPCs and HMCs is large, which may be the molecular basis for DPCs and HMCs playing their distinct roles and conducting cellular interactions in hair follicle development.

### 3.3. Expression Analysis of miRNAs

The miRNA sequencing data showed that 23.9 to 24.69M clean reads of each sample were retained from the raw data. Most of the sequences were distributed across 18–25 nt in length, and the sequences distributed across 22 nt and 23 nt accounted for the highest proportion, which is the typical length distribution of miRNAs (Figure 3a). All the clean reads were mapped to the yak genome sequence, and an average of 23,244,205 (96.0%) clean reads were aligned to the reference genome (Appendix A). After the filtered clean reads were aligned with the Rfam database, cDNA database, and Repbase database using the bowtie_v1.3.1 software, and filtering the rRNA, scRNA, snRNA, tRNA, repeat sequences, and gene fragments, the retained sequences were used for subsequent analysis.

For the identification and annotation of miRNAs, the retained clean reads from above were mapped to the annotated miRNAs in miRbase (version 22.0) (https://www.mirbase.org/, accessed on 29 August 2022) and 71.63~73.45% of the total mapped reads were aligned to the annotated miRNAs (Appendix A). As a result, 757 annotated mature miRNAs were identified, and 2096 novel miRNAs were predicted in all the samples (Appendix A).

The expression levels of annotated and novel miRNAs in each library were calculated and normalized by transcripts per million (TPM) (Appendix A), and their sequences are also listed. PCA showed that the samples in the same group had a closer distance (Figure 3b). A total of 439 significantly different miRNAs were obtained based on the threshold of adjusted *p*-value < 0.05 and |log2FC| > 1, including 276 known miRNAs and 163 predicted novel miRNAs. Among them, 243 miRNAs were upregulated in DPCs, and 196 miRNAs were downregulated in DPCs compared with HMCs (Figure 3c).

### 3.4. GO and KEGG Enrichment Analyses of DEGs

To understand the biological functions of the DEGs involved, GO and KEGG enrichment analyses were performed. GO enrichment results show that the upregulated DEGs in DPCs are mainly enriched in the cell adhesion, collagen fibril organization, extracellular matrix organization, and the GO terms related to transcription factor activity, and this result is consistent with the dermal component characteristics of the DPCs. In addition, the downregulated DEGs were mainly enriched in the GO terms such as wound healing, intracellular signal transduction, and establishment of skin barrier and plasma membrane (Figure 4a), and these functions were similar to epidermal cells. The result also reflected that DPCs were the main cells that synthesized and secreted protein and RNA signals, while HMCs were the response cells on which multiple membrane receptors may exist.

KEGG results of the up- and downregulated genes indicated that 132 signaling pathways were significantly enriched (*p* < 0.05). Some pathways such as Wnt, TGF-β, ECM–receptor interaction, and platelet activation, which were reported to be associated with hair follicle development were enriched in the top 20 KEGG pathways of the upregulated genes in DPCs, whereas multiple pathways related to microbiota and immunity, such as human papillomavirus infection, Influenza A, the IL-17 signaling pathway, and the tight junction were enriched in the downregulated genes of DPCs (Figure 4b).

Given that numerous signaling pathways were significantly enriched, in order to better analyze the overall functional differences between DPCs and HMCs in hair follicle development, the KEGG secondary classification analysis of the significantly enriched signaling pathways was performed. The result showed that the pathways enriched in the endocrine system were dominant in the upregulated genes in DPCs, while more signaling pathways related to the immune system were significantly enriched in the downregulated genes in DPCs compared with HMCs (Figure 4c). These results may be consistent with the function of DPCs which act as the “command center” in hair follicle development to secrete and transmit signals to other nearby hair follicle cells, and the HMCs as a kind of epidermal cell may be functional in the immunity and protective barriers of hair follicles and skin.

Furthermore, it was found that the five pathways including Wnt, Notch, SHH, ECM, and TGF-β, which are the traditional pathways associated with hair follicle development, were all significantly enriched, and these pathways belong to the category of “signaling molecules and interaction” and “signal transduction” in the level 2 pathway classification. To analyze the interaction and signal communication of these pathways in the two types of cells, PPI analysis was used to explore the regulatory network of the DEGs involved in these five pathways. The result showed that the DEGs such as NOTCH3, GSK3B, BMP4, and DNC were the key nodes connecting different signaling pathways, most of the signaling molecules involved in TGFβ and SHH were highly expressed in DPCs, and most of the signaling molecules involved in the NOTCH pathway were highly expressed in HMCs in the network (Figure 4d). This network demonstrated the interaction relationship among the crucial signaling pathways involved in hair follicle development and presented the expression distribution in DPCs or HMCs of the key signaling molecules.

### 3.5. Analysis of the Specifically Expressed Genes in Dermal Papilla Cells

Genes specifically expressed in one type of cell may be crucial for understanding the function of that cell and could be used as marker genes. We analyzed the marker genes of DPCs by combining our previous 10× genomics single-cell transcriptome data in yak hair follicles (the data have not yet been published) with this study’s sequencing data. Theoretically, if the genes were classified as the marker genes of the dermal papilla cell cluster in the single-cell sequencing data, all of them would be upregulated in DPCs in the present study. Based on this, the intersection between the upregulated genes in DPCs of this study and the marker genes in each of the cell clusters of the 10×genomics single-cell sequencing data were analyzed. The result showed that the marker genes of one of the cell clusters in single-cell sequencing data were found to be mostly upregulated in DPCs of the present study (44/52), and there was no intersection with the downregulated genes in DPCs (Figure 5a,b). Moreover, 39 of the 44 genes were found to be differentially expressed during the yak hair follicle cycle by comparative analysis with our previous skin transcriptome data [27], and interestingly, all of these 39 genes were highly expressed in telogen (March) or late telogen (June) (Figure 5c). Meanwhile, one of the marker genes VIM was used to verify the result with histoimmunofluorescence; it was shown that VIM specifically expressed in the dermal papilla of yak hair follicles, and ITGA6, which is highly expressed in HMCs according to the gene expression data, was also detected as a control (Figure 5d). This result is consistent with the previously reported conclusions that dermal papilla cells play the most pivotal role in telogen and late telogen [45], and it also further proves that these genes are the markers of DPCs.

### 3.6. Analysis of the miRNA Function and the Interaction Network between miRNAs and Their Target Genes

In order to excavate the miRNAs that may play roles in cellar communication or hair follicle development, here, we focused on the differentially expressed miRNAs with high expression levels or those relatively specifically expressed in DPCs or HMCs by setting the threshold to the average TPM > 2 and |log2FoldChange| > 2. As a result, a total of 123 differentially expressed miRNAs were screened, including 79 upregulated miRNAs and 44 downregulated miRNAs in DPCs (Figure 6a). Multiple miRNAs among them were reported to be associated with hair follicle development, such as miR-143, miR-214, miR-125b, miR-31, and miR-200, and it could be found that the miRNAs are highly expressed in a particular cell type in the form of an miRNA family; for example, the members in miR-199 family are highly expressed in DPCs, while those of the miR-200 family are highly expressed in HMCs (Table 1).

Target genes of the 123 miRNAs were predicted by TargetScan and miRanda. To better investigate the function of these miRNAs in the process of hair follicle development, the target genes were conjointly analyzed with the differentially expressed genes during yak hair follicle cycle in our previous study [27] and 1145 target genes were identified. GO and KEGG enrichment analyses of these target genes were performed. It is worth noting that most of the enriched GO terms of the target genes are similar to the enrichment result of the upregulated mRNAs in DPCs (Figure 6b and Figure 4a). For example, the GO terms including extracellular matrix organization, collagen fibril organization, cell adhesion, collagen-containing extracellular matrix, and extracellular space were enriched, which may be due to the screening of more upregulated miRNAs in DPCs; this also means that the miRNAs play important roles in the function of DPCs. In addition, the KEGG pathways such as ECM–receptor interaction, focal adhesion, melanogenesis, and platelet activation associated with hair follicle development were enriched. Meanwhile, the KEGG result indicated that the pathways including the calcium signaling pathway, protein digestion and absorption, and axon guidance may be involved in miRNA-mediated regulation of HF development (Figure 6c).

Then, the regulatory network of the 123 miRNAs with their target genes was analyzed using the CytoScape_v3.6 software (Figure 6d). These target genes were obtained by the intersection of the above screened 1145 targets with the DEGs between DPCs and HMCs, and numerous genes, including SFRP4, FGF22, BMP4, PDGFRA, GATA3, and TGFB3, related to HF development were shown to be the target genes of the screened miRNAs (Figure 6d), suggesting that the regulation of these miRNAs in HF development may involve these target genes.

### 3.7. Verification of the Sequencing Data by qPCR

To verify the reliability of the sequencing results, several differentially expressed mRNAs and miRNAs associated with hair follicle development were selected to detect their expression pattern using qPCR (Figure 7a). The results were in concordance with the RNA-seq data, indicating the sequencing results were reliable (Figure 7b). For the miRNAs, to further investigate whether there were miRNAs that transferred from one cell type to another, rather than being endogenous and playing a cellar communication role, five miRNAs were selected to detect their pri-miRNA expression levels in both cell types. The result showed that the expression trend of all the detected pri-miRNAs was consistent with their mature miRNAs (Figure 7c), suggesting that the differentially expressed miRNAs were differentially expressed when they were transcribed as pri-miRNAs. The pri-miR-200 and pri-miR-205 were not detected in DPCs (Figure 7d), suggesting it was possible that miR-200 and miR-205 in DPCs were delivered by HMCs, which may play a role in cellular communication.

Finally, a summary diagram is used to present the expression distribution and interaction relationship of signal molecules involved in key signaling pathways and some miRNAs related to hair follicle development in the hair bulb according to the sequencing data and the analysis of above sections (Figure 8), which will provide useful clues to understanding the complex interactions of these crucial signaling molecules and their regulatory mechanisms in hair follicle development.

## 4. Discussion

A hair follicle is a mini-organ with high cellular heterogeneity composed of a variety of cell types belonging to epidermal cell lineage and dermal cell lineage [15]. The regular development during the hair follicle cycle depends on the signal exchange between the epidermis and dermis, in which the interaction between dermal papilla cells (DPCs) and the surrounding epidermal hair matrix cells (HMCs) is crucial for the growth and development of hair follicles, elaborately regulating the growth and shedding of the hair shaft. Previous studies on hair follicle cells mainly focused on the gene expression characteristics or gene functions of the DPCs [22,56], and few studies have systematically explored the molecular differences between DPCs and HMCs to decipher the molecular biological basis of their interaction. Here, we investigated the differentially expressed mRNAs and miRNAs between DPCs and HMCs by isolating and culturing DPCs and HMCs of yak HFs. Although the method is simple, this study uncovered a great deal of information about the molecular biological differences between DPCs and HMCs as well as the dermis and epidermis.

There was a significant morphological difference between cultured DPCs and HMCs. DPCs are mainly triangular or multi-synaptic phenotypes similar to dermal fibroblasts. The cultured HMCs are cobblestone or paving-stone shaped. The morphological characteristics of DPCs and HMCs isolated in this study were consistent with those previously reported [42,57]. A total of 4600 differentially expressed mRNAs were identified, of which 2265 were upregulated and 2335 were downregulated in DPCs according to analysis of the RNA sequencing data. To understand the functional differences of DPCs and HMCs in HF development, GO and KEGG enrichment analysis of the DEGs were performed. GO enrichment analysis showed that GO items including cell adhesion, collagen fibril organization, extracellular matrix (ECM) organization, basement membrane, and DNA-binding transcription factor activity were mainly enriched in the upregulated genes of DPCs, which is consistent with the extracellular matrix of DPCs being rich in basement membrane proteins and collagens [9,58]. Moreover, the ECM is also crucial for the communication between hair follicle cells. The interactions of ECM molecules between hair follicle cells not only play an important role in maintaining cell-specific phenotypes and the niches of individual cells but are also closely related to the physiological and pathological microenvironment changes of hair follicles [59]. It could be found that the ECM is crucial to the structure and function of DPCs, which also reflects the important role of DPCs in hair follicle development. In addition, the GO items including wound healing, intercellular signal transduction, establishment of skin barrier, keratinization, and plasma membrane were enriched in the downregulated genes in DPCs compared with HMCs, which is in accordance with HMCs being epidermal cells and implies that numerous membrane receptors may exist in the HMCs to receive signals transmitted by DPCs during HF development [60]. KEGG enrichment analysis showed that the signaling pathways associated with HF development including platelet activation, ECM–receptor interaction, TGF-β, and Wnt were significantly enriched in the upregulated genes of DPCs [12,61,62,63]. In the KEGG enrichment result of downregulated genes in DPCs, multiple pathways associated with disease, microbiota, and immunity such as rheumatoid arthritis, human papillomavirus infection, Influenza A, and the IL-17 signaling pathway were enriched. It was reported that the pathways related to microbiota and immune and tight junction are all important components of performing skin barrier functions [64], suggesting that the epidermal cells play a more important role in establishment of the skin barrier compared with dermal cells in HFs and may be more likely to cause disease. In addition, it was reported that the microbiota in HFs can affect hair follicles by releasing cytokines and chemokines, thus affecting the growth of HFs [65].

Furthermore, from an overall perspective, the level 2 pathway classification results showed that the pathways enriched in the endocrine system were dominant in the highly expressed genes in DPCs, while more signaling pathways related to the immune system were significantly enriched in the highly expressed genes in HMCs. This result is consistent with the notion that DPCs act as the “command center” in HF development to secrete and transmit signals to other nearby cells, while HMCs are epidermal cells that play important roles in the function of immunity and the protective barrier of HFs.

Five pathways, including WNT, ECM, TGF-β, NOTCH, and SHH, are the traditional pathways that have been reported to be closely related to HF development [62,66], but their molecular interaction in HF development is complex and is yet to be elucidated. In this study, all of these five signaling pathways were significantly enriched by the enrichment analysis of DEGs, and PPI analysis of the DEGs involved in these pathways showed that the five pathways can be linked by several key signaling nodes. Moreover, most of the DEGs in the regulatory network involved in TGFβ and SHH pathways were found to be highly expressed in DPCs, and most of the signaling molecules involved in the NOTCH pathway were highly expressed in HMCs, which was consistent with the previous study [13,67]. The analysis results provide pivotal clues for the interaction relationship between these different pathways in HF development.

In order to obtain a series of accurate marker genes of DPCs of yak through this study, 39 marker genes of DPCs were identified by combining the present data with our previous 10× genomics single-cell transcriptome data in yak hair follicles. It was found that all of these marker genes were highly expressed in telogen (March) or late telogen (June) [27], which was consistent with the previously reported conclusions that dermal papilla cells play the most pivotal role in telogen and late telogen [45] and indicated that the expression changes of some DEGs during the hair follicle cycle were caused by the number changes of the distinct hair follicle cells.

Studies have shown that miRNAs, as a class of highly conserved non-coding small RNAs, are involved in the regulation of HF development [68,69]. Recently, miRNAs have been found to play an important role in the signal communication between different cells. For example, exosomes, the important carriers mediating signal communication between cells, were reported to contain a large number of miRNAs [70]. The exosome miRNAs of DPCs have also been studied [22]. Given that there is a close interaction between HMCs and DPCs during HF development, we supposed that miRNAs mediated cellular communication between DPCs and HMCs, and differential expression may exist among these miRNAs. We therefore detected the differentially expressed miRNAs between DPCs and HMCs. It was also helpful to reveal the cell-specific miRNAs in different cell types though the present study [71]. To achieve the research purposes, 79 up- and 44 downregulated miRNAs in DPCs compared with HMCs were screened from the significantly differentially expressed miRNAs by setting the threshold at an average TPM > 2, |log2FoldChange| > 2. Among these miRNAs, multiple miRNAs have been reported to be involved in the regulation of HF development, such as miR-143 [49], miR-214 [55], miR-125b [72], miR-31 [48], and the miR-200 family [46]. Previous studies found that the miR-200 family, miR-141, and miR-429 are highly expressed in the epidermis of skin tissues, and miR-31 was mainly expressed in hair matrix cells [48], while the miR-199 family was specifically expressed in HFs containing dermis [73], which is consistent with this study’s finding that the former is highly expressed in HMCs and the expression level of the miR-199 family is higher in DPCs. In addition, in the study of DPC exosomes it was reported that the miR-200 family was specifically expressed in the exosomes of DPCs, while the expression level of miR-200 is lower in dermal papilla cells [22], which is in accordance with the low expression of the miR-200 family in DPCs and high expression in HMCs in the present study. GO and KEGG enrichment analysis of the miRNA target genes showed that the KEGG pathways related to HF development, including ECM–receptor interaction, focal adhesion, melanogenesis, and platelet activation, were significantly enriched. It is worth noting that most of the enriched GO terms are similar to the enrichment result of the highly expressed genes in DPCs, which may be due to the screening of more upregulated miRNAs in DPCs and means that the miRNAs play an important role in the function of DPCs. Finally, we found that the differentially expressed miRNAs were differentially expressed when they were transcribed as pri-miRNAs, and the expression trend was consistent with mature miRNAs, suggesting that the biological process mediated by miRNAs is complex and may involve the transcriptional and post-transcriptional processes of miRNAs.

## 5. Conclusions

This study first investigated the mRNA and miRNA expression difference between DPCs and HMCs of yak HFs. In addition, numerous specifically expressed mRNAs and miRNAs were identified, which indicated that some differentially expressed molecules during different stages of HF development may be caused by the number changes of distinct cell types. This study provides systematic and insightful information for the molecular basis of the interaction between DPCs and HMCs of yak HFs, and will effectively promote the mining and accurate location of related key genes and miRNAs in HFs of yak.

## Figures and Tables

**Figure 1 cells-11-03985-f001:**
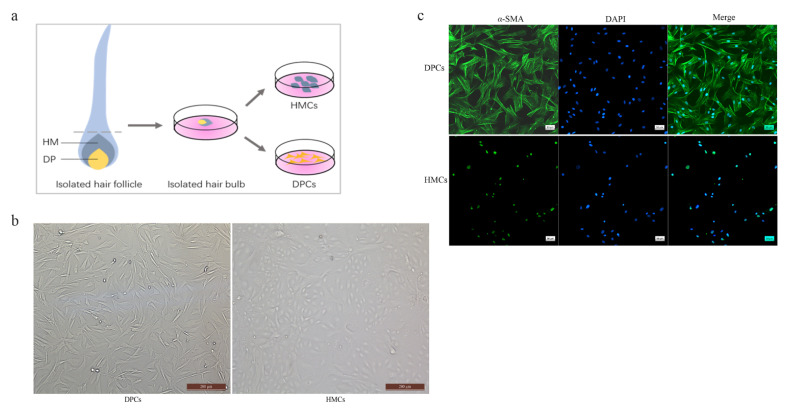
Isolation and identification of DPCs and HMCs from yak. (**a**) Schematic diagram of micro-dissociated hair bulbs from isolated HF and the isolation of DPCs and HMCs. (**b**) The cultured DPCs and HMCs after isolation and purification. (**c**) DPCs were verified by immunofluorescence (IF) of α-SMA. Green fluorescence indicated the expression of α-SMA. The nucleus was stained with DAPI in blue.

**Figure 2 cells-11-03985-f002:**
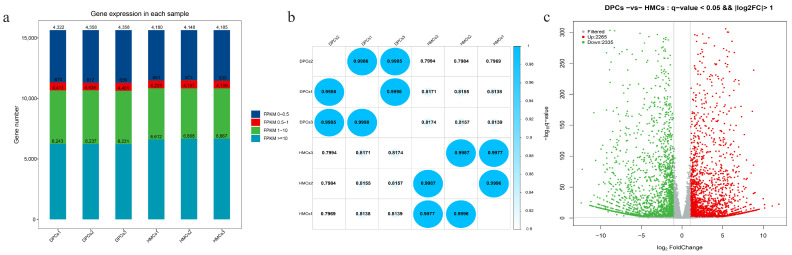
Expression analysis of the mRNA in DPCs and HMCs. (**a**) The expression distribution of FPKM was presented by staking histogram. (**b**) Correlation coefficient between all samples. (**c**) Volcano plot of differentially expressed mRNAs.

**Figure 3 cells-11-03985-f003:**
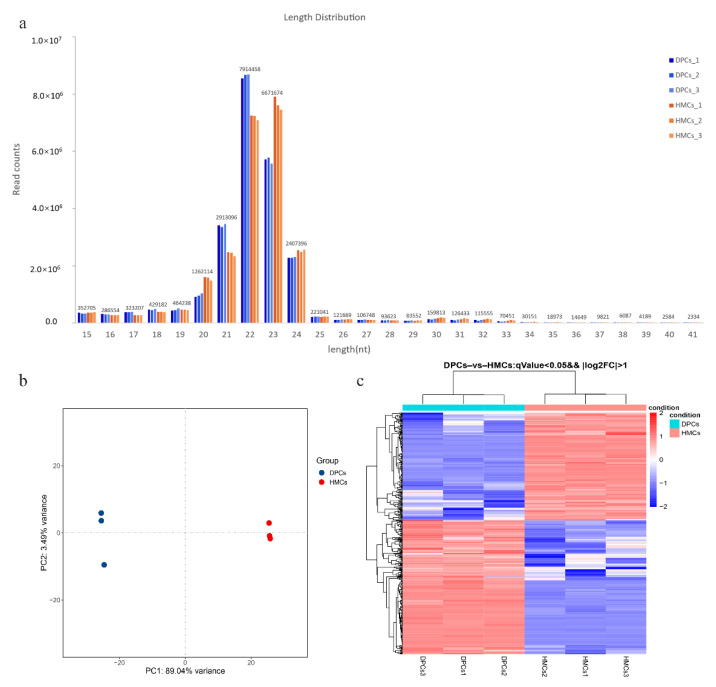
Sequence length distribution and expression analysis of miRNAs. (**a**) Sequence length distribution of the sRNA filtered clean reads. (**b**) PCA analysis based on the expression of miRNAs. (**c**) Heat map clustering of differentially expressed miRNAs.

**Figure 4 cells-11-03985-f004:**
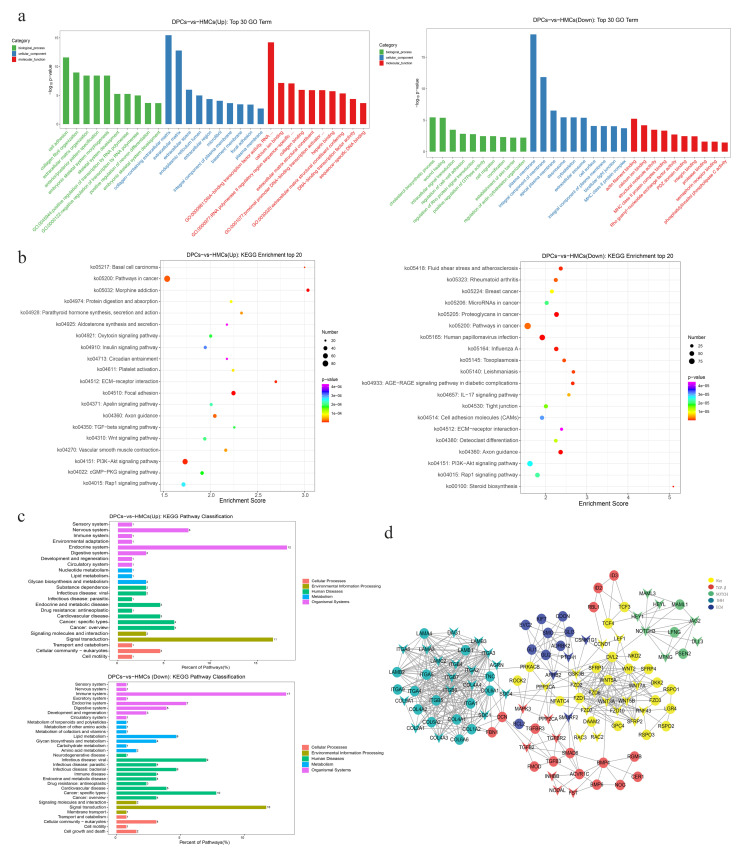
GO and KEGG enrichment analyses of DEGs between DPCs and HMCs. (**a**) GO enrichment analysis results of DEGs, left: top 30 GO terms enriched in upregulated genes (DPCs-vs-HMCs); right: top 30 GO terms enriched in downregulated genes (DPCs-vs-HMCs). (**b**) KEGG enrichment analysis results of DEGs, left: top 20 KEGG pathways enriched in upregulated genes (DPCs-vs-HMCs); right: top 20 KEGG pathways enriched in downregulated genes (DPCs-vs-HMCs). (**c**) Analysis and statistics of level 2 KEGG classification of the significantly enriched pathways; the numbers at the top of the bar chart represents the count of pathways that are significantly enriched. (**d**) PPI network of DEGs involved in the pathways including Wnt, Notch, SHH, and TGF-βand ECM-receptor interaction. The triangle represents downregulation in DPCs and the circle represents upregulation in DPCs. Different colors represent different signaling pathways.

**Figure 5 cells-11-03985-f005:**
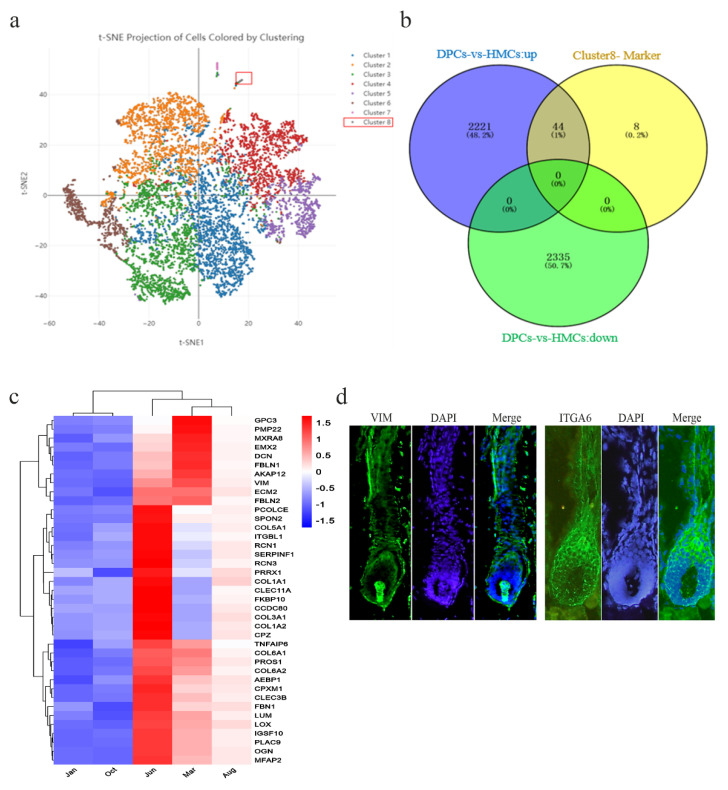
Analysis of the marker genes in DPCs. (**a**) Cell-clustering information of t-SNE clustering analysis of our previous 10× genomics single-cell sequencing data. (**b**) Venn diagram of the marker genes of cluster 8 in the single-cell sequencing data with the up- and downregulated genes in DPCs of the present study. (**c**) Heatmap of the identified 39 marker genes of DPCs during the hair follicle cycle of yak. Jan indicates catagen; Mar and Jun indicate telogen and late telogen, respectively; and Aug and Oct represent anagen during the yak hair follicle cycle. (**d**) Detection of the expression of VIM and ITGA6 in the yak hair follicle, showing that VIM was highly expressed in dermal papilla, and ITGA6 was mainly expressed in the hair matrix. Green fluorescence indicated the expression of the interest protein, and the nucleus was stained with DAPI in blue.

**Figure 6 cells-11-03985-f006:**
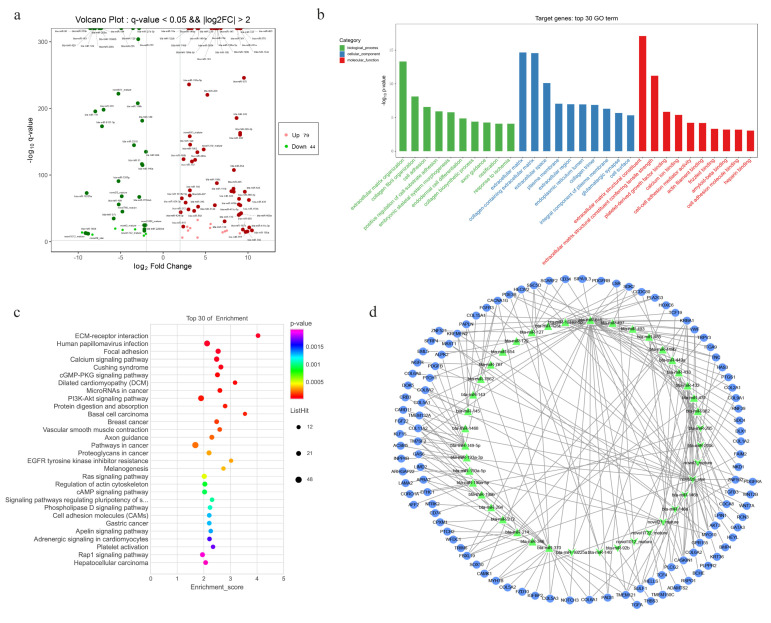
Functional analysis of miRNAs and the interaction network of miRNAs and their potential target genes. (**a**) Volcano plot of the differentially expressed miRNAs screened by setting the threshold to the average TPM > 2 and |log2FoldChange| > 2. (**b**) The top 30 GO terms enriched by the miRNA target genes. (**c**) The top 30 KEGG pathways enriched by the miRNA target genes. (**d**) Network of differentially expressed miRNAs and their potential target genes; these targets are differentially expressed during the hair follicle cycle of yak as well as between DPCs and HMCs. Blue circles represent target genes, the green triangle represents upregulated miRNAs in DPCs, and the green V shape represents downregulated miRNAs in DPCs compared with HMCs.

**Figure 7 cells-11-03985-f007:**
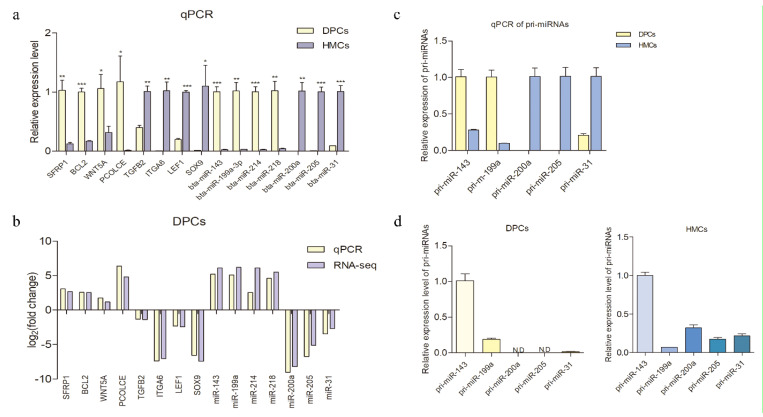
Sequencing data validated by qPCR and the expression trend of the pri-miRNAs was consistent with their mature miRNAs. (**a**) The qPCR was used to detect the expression level of differently expressed mRNAs and miRNAs. (**b**) Comparison analysis of the expression pattern of the sequencing data and qPCR data. Log 2 (fold change) > 0 indicates the transcripts were upregulated in DPCs compared with HMCs. Log 2 (fold change) < 0 indicates the transcripts were downregulated in DPCs compared with HMCs. (**c**) The expression levels of pri-miRNAs of the selected miRNAs in DPCs and HMCs were detected by qPCR. (**d**) The expression levels of pri-miRNAs in DPCs and HMCs are presented, respectively, based on qPCR results; N.D., not detected. The mRNAs and pri-miRNAs were quantified relative to the expression level of *GAPDH*, and miRNA expression was quantified relative to U6. The qPCR data are expressed as the mean ± SEM (*n* = 3). * *p* < 0.05, ** *p* < 0.01, *** *p* < 0.001.

**Figure 8 cells-11-03985-f008:**
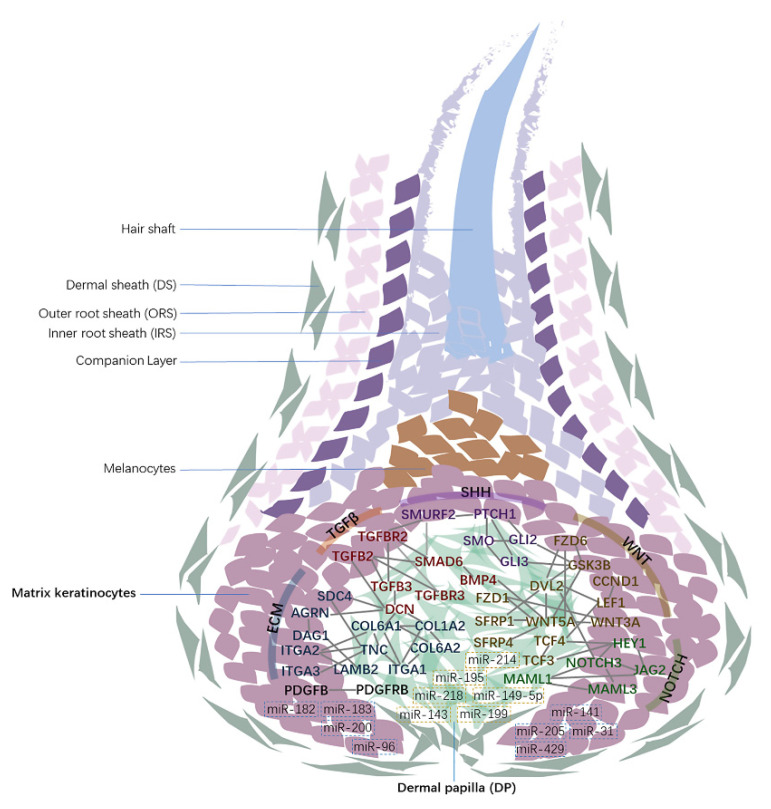
The localization of key differentially expressed mRNAs and miRNAs in DPCs and HMCs and their interaction. The interaction network was constructed with PPI analysis; different colors of text in the network indicate different signaling pathways (Figure 4d).

**Table 1 cells-11-03985-t001:** miRNAs reported to be associated with hair follicle development.

miRNAs	Regulation (DPCs-vs-HMCs)	Function
bta-miR-200a	Down	Regulating cell proliferation in hair morphogenesis [46]
bta-miR-200b	Down	Regulating cell proliferation in hair morphogenesis [46]
bta-miR-200c	Down	Regulating cell proliferation in hair morphogenesis [46]
bta-miR-141	Down	Regulating cell proliferation in hair morphogenesis [46]
bta-miR-429	Down	Regulating cell proliferation in hair morphogenesis [46]
bta-miR-205	Down	Highly expressed in skin stem cells [47]
bta-miR-31	Down	Regulating the expression of keratins [48]
bta-miR-143	Up	Regulating proliferation of dermal papilla cells [49]
bta-miR-149-5p	Up	Promoting differentiation of hair follicle stem cell [50]
bta-miR-195	Up	Hair inductive miRNA [51]
bta-miR-199a-3p	Up	Highly expressed in hair follicles [52]
bta-miR-199a-5p	Up	Inhibited TGF-β2 expression in fibroblasts [53]
bta-miR-199b	Up	Expressed primarily in goat skin [54]
bta-miR-214	Up	Regulating skin morphogenesis and HF cycling [55]
bta-miR-218	Up	Promoting the development of hair follicles [24].

## Data Availability

RNA-seq files (Fastq data and quantitation matrix) were deposited in the GEO database. The access number is GSE214785 and is scheduled to be released on 4 October 2023.

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
