# Peer review of "Comparative Analysis of mRNA and miRNA Expression between Dermal Papilla Cells and Hair Matrix Cells of Hair Follicles in Yak"

_cells, 2022, doi:10.3390/cells11243985_

Round 1

Reviewer 1 Report

The manusctipr is scienifically sound, it has a merit and good idea but you have to send it to native English speaker, because it needs extensive grammar corrections. I started to read it and there was not a single line which was written properly.

In the present form it is not acceptable for publication. Please change the grammatics carefully and after this step I can review the paper again.

I have listed some comments, but due to large amount of changes needed, I was not able to carefully inspect the whole manuscript.

The abstract is poorly written, some sentences are too general and lofty, others are grammatically wrong. Some examples:

Line 20 – „were detected” – I’d rather write „were characterized or described”

Line 21 – „their interaction between DPCs and HMCs” – well, you did not study miRNA/mRNA intereactions between the cells but within DPCs and HMCs

Line 22 – „significant molecular differences” – I’d suggest to delete „molecular”

Lines 25 and 26 – „up-regulated genes” and „down-regulated genes” – this is laboratory Jargon, please change that accordingly

Line 27 – ” identified by combining analysis” – this is not gramatically correct

 Line 28 – please write down the numer 123 in words

Sentences starrting with „The expression and 21 functional analysis of mRNA…” and „In conclusions, a systematic molecular difference…” are really strange, too general and exuberant – please change that

Introduction:

  Line 36 „that is continual” – maybe better „that is characterized with continual”

 Line 37 „consist” instead of „consisted”

Sentence starting with „Dermal and epidermal..” is not grammatically correct

Line 39 – communications are important, not is

Line 41 – prompts instead of prompted

  Line 42 – are is missing before “derived”

 Line 47 – similarly, instead of “similarity”

Line 48 „begin to apoptosis” – this is not grammatically correct

Line 54 – please explain “Shh” abbreviation

 Line 55 “was reported to be coordinated the development” is not grammatically correct

Reviewer 2 Report

In this study, the authors have made a comparison of mRNAs and miRNAs expression between dermal papilla cells and hair matrix cells of hair follicles in yak. This is an excellent piece of work. The methods are clearly written, the results are well presented, and the discussion is appropriate. I see no obstacle in publishing the work. Information obtained in this study would definitely enable researchers to get better insights into the roles of microRNAs during hair follicle development. I only have some minor comments:

1) Authors should mention how many libraries have been prepared for small RNA sequencing.

2) How many biological and technical replicates have been used in qPCR?

3) I suggest authors to cite the following article

Paul, S., Licona-Vázquez, I., Serrano-Cano, F.I., Frías-Reid, N., Pacheco-Dorantes, C., Pathak, S., Chakraborty, S. and Srivastava, A., 2021. Current insight into the functions of microRNAs in common human hair loss disorders: A mini review. Human cell, 34(4), pp.1040-1050.

Author Response

Response to Reviewer 2 Comments

Manuscript ID: cells-2002768

Title: “Comparative analysis of mRNAs and miRNAs expression between dermal papilla cells and hair matrix cells of hair follicle in yak”

In this study, the authors have made a comparison of mRNAs and miRNAs expression between dermal papilla cells and hair matrix cells of hair follicles in yak. This is an excellent piece of work. The methods are clearly written, the results are well presented, and the discussion is appropriate. I see no obstacle in publishing the work. Information obtained in this study would definitely enable researchers to get better insights into the roles of microRNAs during hair follicle development. I only have some minor comments:

Point 1: Authors should mention how many libraries have been prepared for small RNA sequencing.

Response1: Thanks very much for your affirmation to my manuscript. A total of six small RNA libraries from DPCs (n=3) and HMCs (n=3) were constructed for small RNA sequencing, and the relevant description was added in the methods section of the revised manuscript.

 Point 2: How many biological and technical replicates have been used in qPCR?

Response2: There were three biological replicates and three technical replicates in qPCR, and it is mentioned in the figure note of qPCR: “The qPCR data are expressed as the mean ± SEM(n=3)”.

 Point 3: I suggest authors to cite the following article

Paul, S., Licona-Vázquez, I., Serrano-Cano, F.I., Frías-Reid, N., Pacheco-Dorantes, C., Pathak, S., Chakraborty, S. and Srivastava, A., 2021. Current insight into the functions of microRNAs in common human hair loss disorders: A mini review. Human cell, 34(4), pp.1040-1050.

Response3: I have cited this article in line 87 of the revised manuscript, according to the reviewer’s suggestion.

Round 2

Reviewer 1 Report

Thank you for including my comments, I reccomend to publish the paper in the present form